# Robust Graph Representation Learning via Neural Sparsification

## Abstract

Graph representation learning serves as the core of many important prediction tasks, ranging from product recommendation in online marketing to fraud detection in financial domain. Real-life graphs are usually large with complex local neighborhood, where each node is described by a rich set of features and easily connects to dozens or even hundreds of neighbors. Most existing graph learning techniques rely on neighborhood aggregation, however, the complexity on real-life graphs is usually high, posing non-trivial overfitting risk during model training. In this paper, we present Neural Sparsification (NeuralSparse), a supervised graph sparsification technique that mitigates the overfitting risk by reducing the complexity of input graphs. Our method takes both structural and non-structural information as input, utilizes deep neural networks to parameterize the sparsification process, and optimizes the parameters by feedback signals from downstream tasks. Under the NeuralSparse framework, supervised graph sparsification could seamlessly connect with existing graph neural networks for more robust performance on testing data. Experimental results on both benchmark and private datasets show that, NeuralSparse can effectively improve testing accuracy and bring up to 7.4% improvement when working with existing graph neural networks on node classification tasks.

## 1 Introduction

Representation learning has been in the center of many machine learning tasks on graphs, such as name disambiguation in citation networks (Zhang et al., 2018c), spam detection in social networks (Akoglu et al., 2015), recommendations in online marketing (Ying et al., 2018a), and many others (Hamilton et al., 2017; Li et al., 2018). As a class of models that can simultaneously utilize non-structural (*e.g.*, node and edge features) and structural information in graphs, Graph Neural Networks (GNNs) (Kipf & Welling, 2017; Hamilton et al., 2017; Li et al., 2016) construct effective representations for downstream tasks by iteratively aggregating neighborhood information (Kipf & Welling, 2017; Hamilton et al., 2017). Such methods have demonstrated state-of-the-art performance in classification and prediction tasks on graph data (Veličković et al., 2018; Chen et al., 2018; Xu et al., 2019; Veličković et al., 2019).

Meanwhile, graphs from real-life applications are usually large with complex local neighborhood, where each node has rich features and dozens or even hundreds of neighbors. As shown in Figure 1(a), this subgraph from Transaction dataset (detailed in Section 5.1) consists of 38 nodes (*i.e.*, promising organizations and other organizations) with average node degree 15 and node feature dimension 120. The GNNs are expected to grasp useful patterns from neighboring nodes; however, as representative patterns are diluted by overwhelming information in local neighborhood, graph learning algorithms could be misled by neighborhood aggregation. Such complexity in input graphs poses non-trivial overfitting risk to existing GNN based learning techniques.

While it is straightforward yet expensive (sometimes even impractical) to address this overfitting problem by increasing the number of labeled samples, we investigate a cheaper alternative of reducing input graph complexity by graph sparsification in this work. Graph sparsification (Liu et al., 2018; Zhang & Patone, 2017) aims to find smaller subgraphs from input large graphs that best preserve desired properties. Existing sparsification methods could lead to suboptimal performance for downstream prediction tasks: (1) these methods are unsupervised such that the resulting sparsified

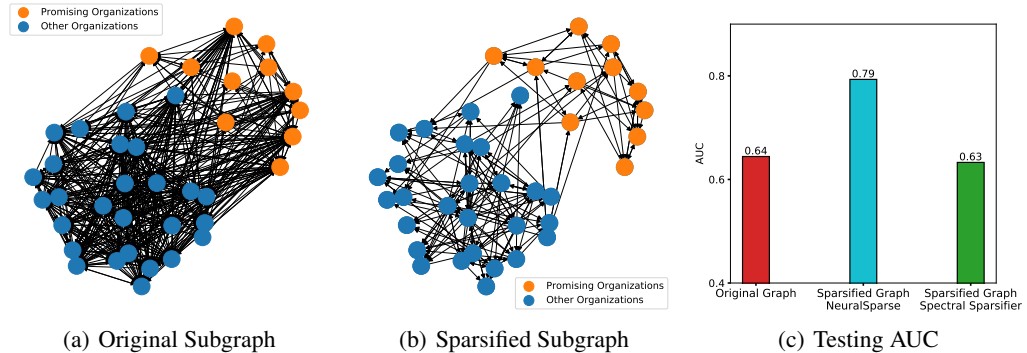

(a) Original Subgraph     (b) Sparsified Subgraph     (c) Testing AUC

Figure 1: A subgraph of 38 organizations from Transaction dataset: (a) The original subgraph sampled from the Transaction dataset, where nodes and edges represent organizations and their transactions, respectively; (b) The sparsified subgraph by NeuralSparse; (c) Testing AUC on identifying promising organizations.

graphs may not favor downstream tasks; and (2) they only consider structural information for sparsification decision, while non-structural information in graphs, such as node/edge features, could have non-trivial impact to the quality of sparsification. Recently, there have been GNN models attempting to sample subgraphs from predefined distributions (Leskovec & Faloutsos, 2006; Adhikari et al., 2018; Hamilton et al., 2017; Chen et al., 2018). As the predefined distributions could be irrelevant to subsequent tasks, the sparsified graphs may miss important information for downstream tasks, leading to suboptimal prediction performance.

**Present work**. We propose Neural Sparsification (NeuralSparse), a general framework that simultaneously learns graph sparsification and graph representation by feedback signals from downstream tasks. The NeuralSparse consists of two major components: sparsification network and GNN. For the sparsification network, we utilize a deep neural network to parameterize the sparsification process: how to select edges from one-hop neighborhood given a fixed budget. In the training phase, the network learns to optimize a sparsification strategy that favors downstream tasks. In the testing phase, the network sparsifies input graphs following the learned strategy, instead of sampling subgraphs from a predefined distribution. Unlike conventional sparsification techniques, our technique takes both structural and non-structural information as input and optimizes the sparsification strategy by feedback from downstream tasks, instead of using (possibly irrelevant) heuristics. For the GNN component, the NeuralSparse feeds the sparsified graphs to a GNN and learns a graph representation for subsequent prediction tasks.

Under the framework of NeuralSparse, we are able to leverage the standard stochastic gradient descent and backpropagation techniques to simultaneously optimize graph sparsification and representation. As shown in Figure 1(b), the graph sparsified by the NeuralSparse has lower complexity with average node degree around 5. As a result (illustrated in Figure 1(c)), the testing classification accuracy on the sparsified graph is improved by 15%, compared with its counterpart in the original input graph, while conventional techniques could not offer competitive sparsification for the classification task.

Experimental results on both public and private datasets show that the NeuralSparse is able to consistently provide improved performance for existing GNNs on node classification tasks, bringing up to 7% improvement.

## 2 RELATED WORK

Our work is related to two lines of research: graph sparsification and graph representation learning.

**Graph sparsification**. The goal of graph sparsification is to find small subgraphs from input large graphs that best preserve desired properties. Existing techniques are mainly unsupervised and deal with simple graphs without node/edge features for preserving predefined graph metrics (Hübler et al., 2008), information propagation traces (Mathioudakis et al., 2011), graph spectrum (Calandriello et al., 2018; Chakeri et al., 2016; Adhikari et al., 2018), node degree distribution (Eden et al.,

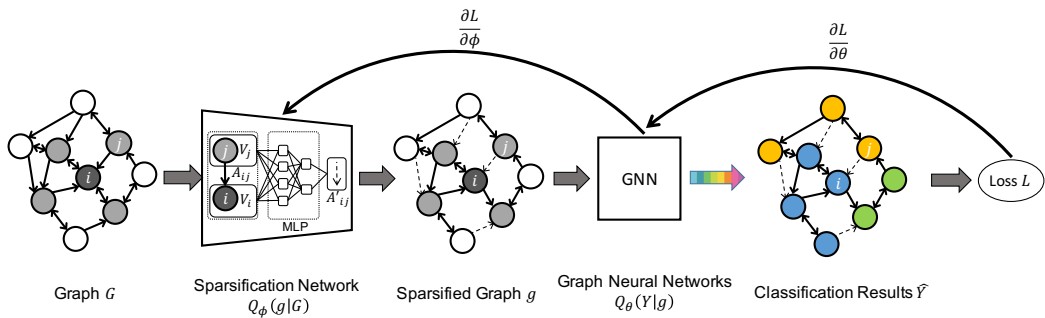

Figure 2: The overview of NeuralSparse

2018; Voudigari et al., 2016), node distance distribution (Leskovec & Faloutsos, 2006), or clustering coefficient (Maiya & Berger-Wolf, 2010). Importance based edge sampling has also been studied in a scenario where we could predefine edge importance (Zhao, 2015; Chen et al., 2018).

Unlike existing methods that mainly work with simple graphs without node/edge features in an unsupervised manner, our method takes node/edge features as parts of input and optimizes graph sparsification by supervision signals from errors made in downstream tasks.

**Graph representation learning**. Graph neural networks (GNNs) are the most popular techniques that enable vector representation learning for large graphs with complex node/edge features. All existing GNNs share a common spirit: extracting local structural features by neighborhood aggregation. Scarselli et al. (2009) explore how to extract multi-hop features by iterative neighborhood aggregation. Inspired by the success of convolutional neural networks, multiple studies (Defferrard et al., 2016; Bruna et al., 2014) investigate how to learn convolutional filters in the graph spectral domain under transductive settings (Zhang et al., 2018b; Zhuang & Ma, 2018). To enable inductive learning, convolutional filters in the graph domain are proposed (Simonovsky & Komodakis, 2017; Niepert et al., 2016; Kipf & Welling, 2017; Veličković et al., 2018; Xu et al., 2018), and a few studies (Hamilton et al., 2017; Lee et al., 2018) explore how to differentiate neighborhood filtering by sequential models. In addition, multiple recent works (Ying et al., 2018b; Xu et al., 2019; Abu-El-Haija et al., 2019) investigate the expressive power of GNNs. Recently, (Franceschi et al., 2019) study how to sample high-quality subgraphs from a space of all possible graphs of a complete graph so that the sampled graphs enhance the prediction power in downstream learning tasks. In particular, the proposed method only focus on transductive tasks.

Our work contributes from a unique angle: by reducing the noise from input graphs, our technique can further boost testing performance of existing GNNs.

## 3 PROPOSED METHOD: NEURALSPARSE

In this section, we introduce the core idea of our method. We start with the notations that are frequently used in this paper. We then describe the theoretical justification behind NeuralSparse and our architecture to tackle the supervised node classification problem.

**Notations**. In this paper, we represent an input graph of $n$ nodes as $G = (V, E, \mathbf{A})$: (1) $V \in \mathbb{R}^{n \times d_n}$ includes node features with dimensionality $d_n$; (2) $E \in \mathbb{R}^{n \times n}$ is a binary matrix where $E(u, v) = 1$ if there is an edge between node $u$ and node $v$; (3) $\mathbf{A} \in \mathbb{R}^{n \times n \times d_e}$ encodes input edge features of dimensionality $d_e$. In addition, we use $Y$ to denote the prediction target in downstream tasks (*e.g.,* $Y \in \mathbb{R}^{n \times d_l}$ if we are dealing with a node classification problem with $d_l$ classes).

**Theoretical justification**. From the perspective of statistical learning, the key of a defined prediction task is to learn $P(Y \mid G)$, where $Y$ is the prediction target and $G$ is an input graph. Instead of directly working with original graphs, we would like to leverage sparsified subgraphs to mitigate overfitting risks. In other words, we are interested in the following variant,

$$P(Y \mid G) \approx \sum_{g \in \mathbb{S}_G} P(Y \mid g)P(g \mid G), \tag{1}$$

where $g$ is a sparsified subgraph, and $\mathbb{S}_G$ is a class of sparsified subgraphs of $G$.

In general, because of the combinatorial complexity in graphs, it is intractable to enumerate all possible $g$ as well as estimate the exact values of $P(Y \mid g)$ and $P(g \mid G)$. Therefore, we approximate the distributions by tractable functions,

$$\sum_{g \in \mathbb{S}_G} P(Y \mid g) P(g \mid G) \approx \sum_{g \in \mathbb{S}_G} Q_\theta(Y \mid g) Q_\phi(g \mid G) \tag{2}$$

where $Q_\theta$ and $Q_\phi$ are approximation functions for $P(Y \mid g)$ and $P(g \mid G)$ parameterized by $\theta$ and $\phi$, respectively.

Moreover, to make the above graph sparsification process differentiable, we employ reparameterization tricks (Jang et al., 2017) to make $Q_\phi(g \mid G)$ directly generate differentiable samples, such that

$$\sum_{g \in \mathbb{S}_G} Q_\theta(Y \mid g) Q_\phi(g \mid G) \propto \sum_{g' \sim Q_\phi(g|G)} Q_\theta(Y \mid g') \tag{3}$$

where $g' \sim Q_\phi(g \mid G)$ means $g'$ is a random sample drawn from $Q_\phi(g \mid G)$.

To this end, the key is how to find appropriate approximation functions $Q_\phi(g \mid G)$ and $Q_\theta(Y \mid g)$.

**Architecture**. In this paper, we propose Neural Sparsification (NeuralSparse) to implement the theoretical framework discussed in Equation 3. As shown in Figure 2, NeuralSparse consists of two major components: sparsification network and GNNs.

- The sparsification network is a multi-layer neural network that implements $Q_\phi(g \mid G)$: Taking $G$ as input, it generates a random sparsified subgraph of $G$ drawn from a learned distribution.

- GNNs implement $Q_\theta(Y \mid g)$ that takes a sparsified subgraph as input, extracts node representations, and makes predictions for downstream tasks.

---

**Algorithm 1** Training algorithm for NeuralSparse

---

**Input:** graph $G = (V, E, \mathbf{A})$, integer $l$, and training labels $Y$.
 1: **while** stop criterion is not met **do**
 2:     Generate sparsified subgraphs $\{g_1, g_2, \cdots, g_l\}$ by sparsification network (Section 4);
 3:     Produce prediction $\{\hat{Y}_1, \hat{Y}_2, \cdots, \hat{Y}_l\}$ by feeding $\{g_1, g_2, \cdots, g_l\}$ into GNNs;
 4:     Calculate loss function $J$;
 5:     Update $\phi$ and $\theta$ by descending $J$
 6: **end while**

---

As the sparsified subgraph samples are differentiable, the two components can be jointly trained using gradient descent based backpropagation techniques from a supervised loss function, as illustrated in Algorithm 1. While the GNNs have been widely investigated in recent works (Kipf & Welling, 2017; Hamilton et al., 2017; Veličković et al., 2018), we focus on the practical implementation for sparsification network in the remaining of this paper.

## 4   SPARSIFICATION NETWORK

Following the theory discussed above, the goal of sparsification network is to generate sparsified subgraphs for input graphs, serving as the approximation function $Q_\phi(g \mid G)$. Therefore, we need to answer the following three questions in sparsification network. **i**). What is $\mathbb{S}_G$ in Equation 1, the class of subgraphs we focus on? **ii**). How to sample sparsified subgraphs? **iii**). How to make sparsified subgraph sampling process differentiable for the end-to-end training? In the following, we address the questions one by one.

$k$**-neighbor subgraphs**. We focus on $k$-neighbor subgraphs for $\mathbb{S}_G$ (Sadhanala et al., 2016): Given an input graph, a $k$-neighbor subgraph shares the same set of nodes with the input graph, and each node in the subgraph can select no more than $k$ edges from its one-hop neighborhood. Although the concept of sparsification network is not limited to a specific class of subgraphs, we choose $k$-neighbor subgraphs for the following reasons.

- We are able to adjust the estimation on the amount of task-relevant graph data by tuning the hyper-parameter $k$. Intuitively, when $k$ is an under-estimate, the amount of task-relevant graph data accessed by GNNs could be inadequate, leading to inferior performance. When $k$ is an over-estimate, the downstream GNNs may overfit the introduced noise or irrelevant graph data, resulting in sub-optimal performance. It could be difficult to set a golden hyper-parameter that works all time, but one has the freedom to choose the $k$ that is the best fit for a specific task.

- $k$-neighbor subgraphs are friendly to parallel computation. As each node selects its edges independently from its neighborhood, we can utilize tensor operations in existing deep learning frameworks, such as tensorflow (Abadi et al., 2016), to speed up the sparsification process.

**Sampling $k$-neighbor subgraphs**. Given $k$ and an input graph $G = (V, E, \mathbf{A})$, we obtain a $k$-neighbor subgraph by repeatedly sampling edges for each node in the original graph. Without loss of generality, we sketch this sampling process by focusing on a specific node $u$ in graph $G$. Let $\mathbb{N}_u$ be the set of one-hop neighbors of node $u$.

1. $v \sim f_\phi(V(u), V(\mathbb{N}_u), \mathbf{A}(u))$, where $f_\phi(\cdot)$ is a function that generates a one-hop neighbor $v$ from the learned distribution based on node $u$'s attributes, node attributes of $u$'s neighbors $V(\mathbb{N}_u)$, and their edge attributes $\mathbf{A}(u)$. In particular, the learned distribution is encoded by parameters $\phi$.

2. Edge $E(u, v)$ is selected for node $u$.

3. The above two steps are repeated $k$ times.

Note that the above process performs sampling without replacement. Given a node $u$, each of its adjacent edges is selected at most once. Moreover, the sampling function $f_\phi(\cdot)$ is shared among nodes; therefore, the number of parameters $\phi$ is independent of the input graph size.

**Making samples differentiable**. While conventional methods are able to generate discrete samples (Sadhanala et al., 2016), these samples are not differentiable such that it is difficult to utilize them to optimize sample generation. To make samples differentiable, we propose a Gumbel-Softmax based multi-layer neural network to implement the sampling function $f_\phi(\cdot)$ discussed in above.

To make the discussion self-contained, we briefly discuss the idea of Gumbel-Softmax. Gumbel-Softmax is a reparameterization trick used to generate differentiable discrete samples (Jang et al., 2017; Maddison et al., 2017). Under appropriate hyper-parameter settings, Gumbel-Softmax is able to generate continuous vectors that are as "sharp" as one-hot vectors widely used to encode discrete data.

Without loss of generality, we focus on a specific node $u$ in a graph $G = (V, E, \mathbf{A})$. Let $\mathbb{N}_u$ be the set of one-hop neighbors of node $u$. We implement $f_\phi(\cdot)$ as follows.

1. $\forall v \in \mathbb{N}_u$,
$$z_{u,v} = \text{MLP}_\phi(V(u), V(v), \mathbf{A}(u, v)), \tag{4}$$
where $\text{MLP}_\phi$ is a multi-layer neural network with parameters $\phi$.

2. $\forall v \in \mathbb{N}_u$, we employ a softmax function to compute the probability to sample the edge,
$$\pi_{u,v} = \frac{\exp(z_{u,v})}{\sum_{w \in \mathbb{N}_u} \exp(z_{u,w})} \tag{5}$$

3. Using Gumbel-Softmax, we generate differentiable samples
$$x_{u,v} = \frac{\exp((\log(\pi_{u,v}) + \epsilon_v)/\tau)}{\sum_{w \in \mathbb{N}_u} \exp((\log(\pi_{u,w}) + \epsilon_w)/\tau)} \tag{6}$$
where $x_{u,v}$ is a scalar, $\epsilon_v = -\log(-\log(s))$ with $s$ randomly drawn from Uniform$(0, 1)$, and $\tau$ is a hyper-parameter called *temperature* which controls the interpolation between discrete distribution and continuous categorical densities.

Note that when we sample $k$ edges, the computation for $z_{u,v}$ and $\pi_{u,v}$ only needs to be performed once. For the hyper-parameter $\tau$, we discuss how to tune it as follows.

**Discussion on temperature tuning**. The behavior of Gumbel-Softmax is governed by a hyper-parameter $\tau$ called temperature. In general, when $\tau$ is small, the Gumbel-Softmax distribution

resembles the discrete distribution, which induces strong sparsity; however, small $\tau$ also introduces high variance gradient that blocks effective backpropagation. A high value of $\tau$ cannot produce expected sparsification effect. Following the practice in (Jang et al., 2017), we adopt the strategy by starting the training with a high temperature and anneal to a small value with a guided schedule.

**Sparsification algorithm and its complexity**. As shown in Algorithm 2, given hyper-parameter $k$, the sparsification network visits each node's one-hop neighbors $k$ times. Let $m$ be the total number of edges in the graph. The complexity of sampling subgraphs by the sparsification network is $O(km)$. When $k$ is small in practice, the overall complexity is $O(m)$.

---

**Algorithm 2** Sampling subgraphs by sparsification network

---

**Input:** graph $G = (V, E, \mathbf{A})$ and integer $k$.
1: Edge set $\mathbb{H} = \emptyset$
2: **for** $u \in \mathbb{V}$ **do**
3:     **for** $v \in \mathbb{N}_u$ **do**
4:         $z_{u,v} \leftarrow \mathrm{MLP}_\phi(V(u), V(v), \mathbf{A}(u,v))$
5:     **end for**
6:     **for** $v \in \mathbb{N}_u$ **do**
7:         $\pi_{u,v} \leftarrow \exp(z_{u,v})/\sum_{w \in \mathbb{N}_u} \exp(z_{u,w})$
8:     **end for**
9:     **for** $j = 1, \cdots, k$ **do**
10:        **for** $v \in \mathbb{N}_u$ **do**
11:           $x_{u,v} \leftarrow \exp((\log(\pi_{u,v}) + \epsilon_v)/\tau)/\sum_{w \in \mathbb{N}_u} \exp((\log(\pi_{u,w}) + \epsilon_w)/\tau)$
12:        **end for**
13:        Add the edge represented by vector $[x_{u,v}]$ into $\mathbb{H}$
14:     **end for**
15: **end for**

---

**Comparison with multiple related methods**. Unlike GraphSAGE (Hamilton et al., 2017), Fast-GCN (Chen et al., 2018), and AS-GCN (Huang et al., 2018) that incorporate layer-wise node samplers to reduce the complexity of GNNs, NeuralSparse samples subgraphs before applying GNNs. As for the computation complexity, the sparsification in NeuralSparse is more friendly to parallel computation than the layer-conditioned approach in AS-GCN. Compared with GAT (Veličković et al., 2018; Zhang et al., 2018a), the NeuralSparse can produce sparser neighborhood, which effectively mitigates overfitting risks. Unlike LDS (Franceschi et al., 2019), NeuralSparse learns inductive graph sparsification, and its graph sampling is constrained by input graph topology.

## 5 EXPERIMENTAL STUDY

In this section, we evaluate our proposed NeuralSparse on node classification task, including inductive and transductive settings. We demonstrate that NeuralSparse achieves superior classification performance over state-of-the-art GNN models. Moreover, we provide a case study to demonstrate how sparsified subgraphs generated by NeuralSparse could improve classification. The supplementary material contains more detailed experimental information.

### 5.1 DATASETS

We employ five datasets from various domains and conduct node classification task following the settings as described in Hamilton et al. (2017); Kipf & Welling (2017). The dataset statistics are summarized in Table 1.

**Inductive datasets.** We utilize the Reddit and PPI datasets and follow the same setting in Hamilton et al. (2017). The Reddit dataset contains post-to-post graph with word vectors as node features. The node labels represent which community Reddit posts belong to. The protein-protein interaction (PPI) dataset contains graphs corresponding to different human tissues. The node features are positional gene sets, motif gene sets and immunological signatures. The nodes are multi-labeled by gene ontology. The graph in the Transaction dataset contains real transactions between organizations in two years, with the first year for training and the second year for validation/testing. Each node

Table 1: Dataset statistics

| | **Reddit** | **PPI** | **Transaction** | **Cora** | **Citeseer** |
|---|---|---|---|---|---|
| **Task** | Inductive | Inductive | Inductive | Transductive | Transductive |
| **Nodes** | 232,965 | 56,944 | 95,544 | 2,708 | 3,327 |
| **Edges** | 11,606,919 | 818,716 | 963,468 | 5,429 | 4,732 |
| **Features** | 602 | 50 | 120 | 1,433 | 3,703 |
| **Classes** | 41 | 121 | 2 | 7 | 6 |
| **Training Nodes** | 152,410 | 44,906 | 47,772 | 140 | 120 |
| **Validation Nodes** | 23,699 | 6,514 | 9,554 | 500 | 500 |
| **Testing Nodes** | 55,334 | 5,524 | 38,218 | 1,000 | 1,000 |

represents an organization and each edge indicates a transaction between two organizations. Node attributes are side information about the organizations such as account balance, cash reserve, etc. On this dataset, we aim to classify organizations into two categories: *promising* or *others* for investment in near future. The class distribution in the Transaction dataset is highly imbalanced. During the training under inductive setting, algorithms have only access to training nodes' attributes and edges. In the PPI and Transaction datasets, the models have to generalize to completely unseen graphs.

**Transductive datasets.** We use two citation benchmark datasets with transductive experimental setting in Yang et al. (2016); Kipf & Welling (2017). The citation graphs contain nodes corresponding to documents and edges as citations. Node features are the sparse bag-of-words representations of documents and node labels indicate the topic class of the documents. In transductive learning, the training methods have access to all node features and edges, with a limited subset of node labels.

## 5.2 EXPERIMENTAL SETUP

**Baseline models**. We incorporate four state-of-the-art methods as the base GNN components, including GCN (Kipf & Welling, 2017), GraphSAGE (Hamilton et al., 2017), GAT (Veličković et al., 2018), and GIN (Xu et al., 2019). We evaluate our proposed NeuralSparse with sparsification network and each of the four GNNs. Besides, we also implement variants of NeuralSparse by replacing the sparsification network with either the spectral sparsifier (SS, Sadhanala et al., 2016) or the Rank Degree (RD, Voudigari et al., 2016) method.

**Temperature tuning.** We anneal the temperature with the schedule $\tau = \max(0.05, \exp(-rp))$, where $p$ is the training epoch and $r \in 10^{\{-5,-4,-3,-2,-1\}}$. $\tau$ is updated every $N$ steps and $N \in \{50, 100, ..., 500\}$. Compared with MNIST VAE model in Jang et al. (2017), smaller hyperparameter $\tau$ fits NeuralSparse better in practice.

**Metrics.** We evaluate the performance on the transductive datasets with accuracy (Kipf & Welling, 2017). For inductive tasks on the Reddit and PPI datasets, we report micro-averaged F1 scores (Hamilton et al., 2017). Due to the highly imbalanced classes in the Transaction dataset, models are evaluated with AUC value (Huang & Ling, 2005). The results show the average of 10 runs.

## 5.3 CLASSIFICATION PERFORMANCE

Table 2 summarizes classification performance of NeuralSparse and the baseline methods on all datasets. For Reddit, PPI, Transaction, Cora and Citeseer, the hyper-parameter $k$ is set as 30, 15, 10, 5, and 3 respectively. The hyper-parameter $l$ is set as 1 in this experiment. Note that the result of GAT on Reddit is missing due to the out-of-memory error.

Overall, NeuralSparse is able to help GNN techniques achieve competitive generalization performance with sparsified graph data. We make the following observations. (1) Compared with basic GNN models, NeuralSparse can enhance the generalization performance on node classification tasks by utilizing the sparsified subgraphs from sparsification network, especially in the inductive setting. Indeed, large neighborhood size in the original graphs could bring increased chance of introducing noise into the convolutional operations, leading to sub-optimal performance. (2) With different GNN models, the NeuralSparse can consistently achieve comparable or superior performance, which demonstrates NeuralSparse is general and can be applied to multiple classification models.

Table 2: Node classification performance

| Sparsifier | Method | Reddit | PPI | Transaction | Cora | Citeseer |
|---|---|---|---|---|---|---|
| | | Micro-F1 | Micro-F1 | AUC | Accuracy | Accuracy |
| N/A | GCN | $0.922 \pm 0.041$ | $0.532 \pm 0.024$ | $0.564 \pm 0.018$ | $0.810 \pm 0.027$ | $0.694 \pm 0.020$ |
| | GraphSAGE | $0.938 \pm 0.029$ | $0.600 \pm 0.027$ | $0.574 \pm 0.029$ | $0.825 \pm 0.033$ | $0.710 \pm 0.020$ |
| | GAT | - | $0.917 \pm 0.030$ | $0.616 \pm 0.022$ | $0.821 \pm 0.043$ | $0.721 \pm 0.037$ |
| | GIN | $0.928 \pm 0.022$ | $0.703 \pm 0.028$ | $0.607 \pm 0.031$ | $0.816 \pm 0.020$ | $0.709 \pm 0.037$ |
| SS/ RD* | GCN | $0.912 \pm 0.022$ | $0.521 \pm 0.024$ | $0.562 \pm 0.035$ | $0.780 \pm 0.045$ | $0.684 \pm 0.033$ |
| | GraphSAGE | $0.907 \pm 0.018$ | $0.576 \pm 0.022$ | $0.565 \pm 0.042$ | $0.806 \pm 0.032$ | $0.701 \pm 0.027$ |
| | GAT | - | $0.889 \pm 0.034$ | $0.614 \pm 0.044$ | $0.807 \pm 0.047$ | $0.686 \pm 0.034$ |
| | GIN | $0.901 \pm 0.021$ | $0.693 \pm 0.019$ | $0.593 \pm 0.038$ | $0.785 \pm 0.041$ | $0.706 \pm 0.043$ |
| **Neural Sparse** | GCN | $0.946 \pm 0.020$ | $0.600 \pm 0.014$ | $0.610 \pm 0.022$ | $0.821 \pm 0.014$ | $0.715 \pm 0.014$ |
| | GraphSAGE | $\mathbf{0.951} \pm 0.015$ | $0.626 \pm 0.023$ | $0.649 \pm 0.018$ | $0.832 \pm 0.024$ | $0.720 \pm 0.013$ |
| | GAT | - | $\mathbf{0.921} \pm 0.015$ | $\mathbf{0.671} \pm 0.018$ | $\mathbf{0.834} \pm 0.015$ | $\mathbf{0.724} \pm 0.026$ |
| | GIN | $0.937 \pm 0.027$ | $0.744 \pm 0.015$ | $0.634 \pm 0.023$ | $0.824 \pm 0.027$ | $0.719 \pm 0.015$ |

(* Report the better performance with SS or RD)

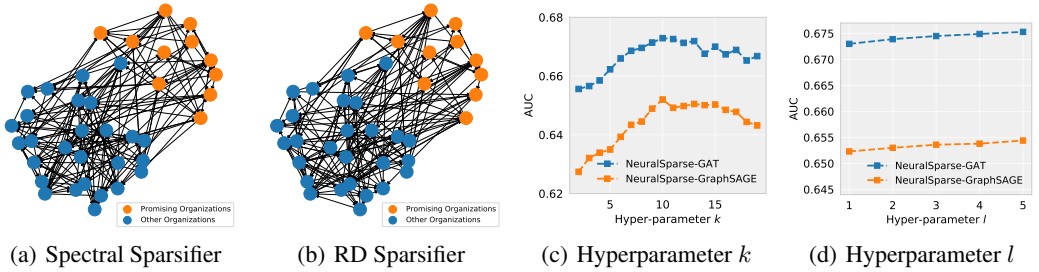

(a) Spectral Sparsifier     (b) RD Sparsifier     (c) Hyperparameter $k$     (d) Hyperparameter $l$

Figure 3: Sparsified subgraphs and performance vs hyper-parameters

(3) In comparison with the two NeuralSparse variants SS-GraphSAGE and RD-GraphSAGE, NeuralSparse outperforms because of the automatically learned graph sparsification with both structural and non-structural information as input.

## 5.4 SENSITIVITY TO HYPER-PARAMETERS AND SPARSIFIED SUBGRAPHS

Figure 3(c) demonstrates how classification performance responds when $k$ increases on the Transaction dataset. There exists an optimal $k$ that delivers the best classification AUC score. When $k$ is small, NeuralSparse can only make use of little relevant structural information in feature aggregation, which leads to inferior performance. When $k$ increases, the aggregation convolution involves more complex neighborhood aggregation with higher chance of overfitting noise data, which negatively impacts the classification performance for unseen testing data. Figure 3(d) shows how hyperparameter $l$ impacts classification performance on the Transaction dataset. When $l$ increases from $1$ to $5$, we observe a relatively small improvement in classification AUC score. As the parameters in the sparsification network are shared by all edges in the graph, the estimation variance from random sampling could already be mitigated to some extent by a number of sampled edges in a sparsified subgraph. Thus, when we increase the number of sparsified subgraphs, the incremental gain could be small.

In Figure 3(a, b), we present the sparsified graphs output by two baseline methods, SS and RD. By comparing the two plots with Figure 1(b), we make the following observations. First, the NeuralSparse sparsified graph tends to select edges that connect nodes of identical labels, which favors the downstream classification task. The observed clustering effect could further boost the confidence of decision making. Second, instead of exploring all the neighbors, we can focus on selected connections/edges in sparsified graphs, which could make it easier for human experts to perform model interpretation and result visualization.

## 6 CONCLUSION

In this paper, we propose Neural Sparsification (NeuralSparse) to address the overfitting issues brought by the complexity in real-life large graphs. NeuralSparse consists of two major components: (1) The sparsification network sparsifies input graphs by sampling edges following a learned distribution; (2) GNNs take sparsified subgraphs as input and extracts node representations for downstream tasks. The two components in NeuralSparse can be jointly trained with supervised loss, gradient descent, and backpropagation techniques. The experimental study on real-life datasets show that the NeuralSparse consistently renders more robust graph representations, and brings up to 7% improvement in accuracy over the state-of-the-art GNN models.

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

## S1    DATASET DETAILS

In this section, we provide additional details about the Transaction datasets in our experiments. The Transaction dataset contains attributed graph that records transaction history between organizations in two years: 2014 and 2015. Each node represents an organization and each directed edge indicates a transaction between two organizations. Node attributes include organization information like account balance, cash research, etc. Under the inductive experimental setting, We use the 47,772 organization data of the year 2014 for training and remaining data are hidden from the model. The 9,554 organizations are used for validation and 38,218 for testing. Validation and testing node sets are from year 2015 and are not connected to the nodes in the training set. Like the PPI dataset, models need to generalize to unseen graph when testing on the Transaction dataset.

## S2    EXPERIMENTAL SETTINGS

In this section, we provide more details about our implementation and experiments in favor of reproducibility.

### S2.1    HARDWARE

All experiment are run on a Linux machine with 16 Intel(R) Xeon(R) CPU (E5-2637 v4 @ 3.50GHz) and 128GB RAM. Some models (e.g. NeuralSparse and GCN) are accelerated by 4 NVIDIA GeForce GTX1080Ti GPU with 11GB RAM.

### S2.2    IMPLEMENTATIONS OF NEURALSPARSE

We implement the proposed NeuralSparse in tensorflow framework for efficient GPU computation. In particular, the multi-layer neural network (Equation 4) in the sparsification network is implemented by two-layer feed-forward neural networks in all experiment, where the hyper-parameter $k$ is searched between 2 and 50 for the optimal performance. We employ cross-entropy to formulate the loss function and apply Adam optimizer for training. The learning rate of Adam optimizer is initially set to be $\alpha = 1.0 \times 10^{-3}$. We initial the weight matrices in the proposed NeuralSparse model with Xavier initialization.

In the following, we detail the network structures of NeuralSparse used on individual datasets. FC($a$, $b$, $f$) means a fully-connected layer with $a$ input neurons and $b$ output neurons activated by function $f$ (none means no activation function is used). GNN($a$, $b$, $f$) means a Graph Neural Network layer with input dimension $a$, output dimension $b$, and activation function $f$. We implement GNN layer with GCN, GraphSAGE, GAT, GIN in the experiments.

**Reddit** The sparsification network runs with: FC(1204, 16, ReLU)-FC(16, 1, Gumbel-Softmax). The structure of GNN is GNN(602, 128, ReLU)-GNN(128, 64, ReLU)-FC(64, 41, softmax).

**PPI** The sparsification network runs with: FC(100, 16, ReLU)-FC(16, 1, Gumbel-Softmax). The structure of GNN is GNN(50, 128, ReLU)-GNN(128, 128, ReLU)-FC(128, 121, softmax).

**Transaction** The sparsification network runs with: FC(243, 16, ReLU)-FC(16, 1, Gumbel-Softmax). The structure of GNN is GNN(121, 128, ReLU)-GNN(128, 32, ReLU)-FC(32, 2, softmax). Note that there is one-dimensional edge attribute indicating the transaction amount in this dataset.

**Cora** The sparsification network runs with: FC(2866, 32, ReLU)-FC(32, 1, Gumbel-Softmax). The structure of GNN is GNN(1433, 128, ReLU)-GNN(128, 64, ReLU)-FC(64, 7, softmax).

**Citeseer** The sparsification network runs with: FC(7406, 64, ReLU)-FC(64, 1, Gumbel-Softmax). The structure of GNN is GNN(3703, 128, ReLU)-GNN(128, 64, ReLU)-FC(64, 6, softmax).

As the spectral sparsification models cannot be jointly trained with subsequent GNN module, the sparsification process is treated as a preprocessing step. For Spectral Sparsifier (SS), $\epsilon$ is set to $0.4$ in all datasets. For the Rank Degree algorithm (RD), we select $1\%$ of nodes as the initial seeds and adopt $\rho \in \{0.1, 0.2, \cdots, 0.8\}$ for the best results.

## S3 Qualitative Edge Sampling Evaluation

In this section, we qualitatively demonstrate the difference by Figure 1(a) original graph, Figure 1(b) NeuralSparse, Figure 3(a) SS, and Figure 3(b) RD. In addition, we provide quantitative analysis in Table S1, where we report the percentage of edges that connect nodes of same class labels in sparsified graphs. Both qualitative and quantitative results suggest a common trend: NeuralSparse prefers to select neighbors with the same labels compared with the baseline methods.

Table S1: Percentage of edges connecting nodes of the same labels

|  | **Reddit** | **PPI** | **Transaction** | **Cora** | **Citeseer** |
|---|---|---|---|---|---|
| Original | 53.1% | 55.0% | 67.3% | 82.2% | 73.1% |
| SS | 50.9% | 52.8% | 62.8% | 79.8% | 75.6% |
| RD | 49.8% | 53.5% | 63.4% | 84.8% | 72.3% |
| **NeuralSparse** | 59.6% | 61.5% | 76.8% | 93.1% | 87.4% |

## S4 Experiment with similar numbers of trainable parameters

In this section, we evaluate the impact brought by reducing the number of parameters in a GNN with NeuralSparse so that the numbers of trainable parameters in a NeuralSparse GNN and an original GNN are similar. In particular, we focus on GCN in this set of experiment. Using the same notation in S2.2, NeuralSparse-GCN-Compact is implemented as follows.

**Reddit**. NeuralSparse-GCN-Compact runs with: FC(1204, 8, ReLU)-FC(8, 1, Gumbel-Softmax) and GCN(602, 112, ReLU)-GCN(112, 64, ReLU)-FC(64, 41, softmax). The total number of trainable parameters is 86,856 in the NeuralSparse-GCN-Compact, while it is 87,872 in the original GCN.

**PPI**. NeuralSparse-GCN-Compact runs with: FC(100, 16, ReLU)-FC(16, 1, Gumbel-Softmax) and GCN(50, 118, ReLU)-GCN(118, 128, ReLU)-FC(128, 121, softmax). The total number of trainable parameters is 38,108 in the NeuralSparse-GCN-Compact, while it is 38,272 in the original GCN.

**Transaction**. NeuralSparse-GCN-Compact runs with: FC(243, 16, ReLU)-FC(16, 1, Gumbel-Softmax) and GCN(121, 100, ReLU)-GCN(100, 32, ReLU)-FC(32, 2, softmax). The total number of trainable parameters is 19,268 in the NeuralSparse-GCN-Compact, while it is 19,648 in the original GCN.

**Cora** NeuralSparse-GCN-Compact runs with: FC(2866, 8, ReLU)-FC(8, 1, Gumbel-Softmax) and GCN(1433, 115, ReLU)-GCN(115, 32, ReLU)-FC(32, 7, softmax). The total number of trainable parameters is 191,635 in the NeuralSparse-GCN-Compact, while it is 192,064 in the original GCN.

**Citeseer** NeuralSparse-GCN-Compact runs with: FC(7406, 32, ReLU)-FC(32, 1, Gumbel-Softmax) and GCN(3703, 64, ReLU)-GCN(64, 32, ReLU)-FC(32, 6, softmax). The total number of trainable parameters is 476,256 in the NeuralSparse-GCN-Compact, while it is 482,560 in the original GCN.

Table S2: Node classification performance with similar numbers of trainable parameters

| Dataset | **Reddit** | **PPI** | **Transaction** | **Cora** | **Citeseer** |
|---|---|---|---|---|---|
| Metrics | Micro-F1 | Micro-F1 | AUC | Accuracy | Accuracy |
| GCN | $0.922 \pm 0.041$ | $0.532 \pm 0.024$ | $0.564 \pm 0.018$ | $0.810 \pm 0.027$ | $0.694 \pm 0.020$ |
| NeuralSparse-GCN | $0.946 \pm 0.020$ | $0.600 \pm 0.014$ | $0.610 \pm 0.022$ | $0.821 \pm 0.014$ | $0.715 \pm 0.014$ |
| NeuralSparse-GCN-Compact | $0.943 \pm 0.018$ | $0.601 \pm 0.021$ | $0.605 \pm 0.013$ | $0.820 \pm 0.012$ | $0.713 \pm 0.009$ |

From the evaluation results shown in Table S2, we draw the following observations. First, both NeuralSparse-GCN and NeuralSparse-GCN-Compact consistently outperform GCN on all the

datasets. Second, compared with NeuralSparse-GCN, NeuralSparse-GCN-Compact achieves comparable prediction accuracy with smaller variance in most cases.

## S5  HOW PERFORMANCE EVOLVES AS HYPER-PARAMETER $k$ CHANGES?

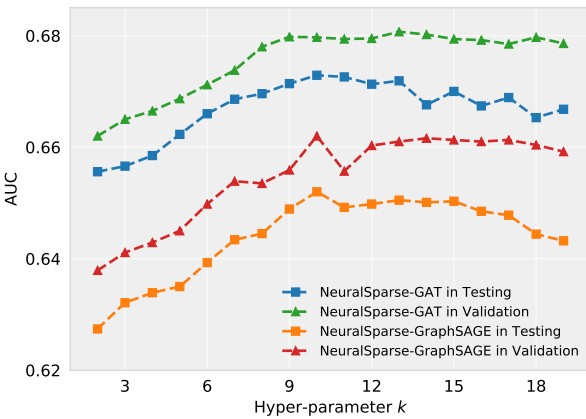

Figure S1: Impact from hyper-parameter $k$ on validation and testing on the Transaction dataset

In this section, we demonstrate how the hyper-parameter $k$ impacts the performance of NeuralSparse-GAT and NeuralSparse-GraphSAGE in both validation and testing on the Transaction dataset. In terms of validation, as shown in Figure S1, the validation performance increases when $k$ ranges from 2 to 10 with more available graph data. After $k$ exceeds 10, the increase in validation performance slows down and turns to be saturated. In terms of testing performance, it shares a similar trend when $k$ ranges from 2 to 10. Meanwhile, the testing performance drops more after $k$ exceeds 10.

## S6  EMPIRICAL COMPARISON BETWEEN NEURALSPARSE AND LDS

### S6.1  EVALUATION RESULTS

In this section, we compare NeuralSparse and LDS (Franceschi et al., 2019) with the datasets in transductive setting. Here, we utilize three ways to prepare the input graphs of Cora and Citeseer datasets.

- Setting A: k-NN graphs (Franceschi et al., 2019). In this setting, the graph structures are completely missing. The input graphs are replaced with k-nearest neighbor graphs initialized from node features. The $k$ in kNN graph is selected from $\{10, 20\}$.
- Setting B: original input graphs of Cora and Citeseer datasets with the same random split as Kipf & Welling (2017).
- Setting C: edge union of original input graphs and k-NN graphs with $k$ fixed as 10.

Table S3: Node classification performance in setting A

|  | Cora(10) | Cora(20) | Citeseer(10) | Citeseer(20) |
|---|---|---|---|---|
| GCN | $0.641 \pm 0.009$ | $0.631 \pm 0.013$ | $0.653 \pm 0.012$ | $0.671 \pm 0.019$ |
| LDS-GCN | $0.715 \pm 0.008$ | $0.703 \pm 0.011$ | $0.691 \pm 0.021$ | $0.715 \pm 0.011$ |
| NeuralSparse-GCN | $0.723 \pm 0.012$ | $0.719 \pm 0.008$ | $0.731 \pm 0.011$ | $0.724 \pm 0.017$ |

Our observation is summarized as follows. In general, NeuralSparse and LDS achieves comparable node classification accuracy. Specifically, NeuralSparse has relatively better performance in Setting

Table S4: Node classification performance in setting B

|  | Cora | Citeseer |
|---|---|---|
| GCN | $0.810 \pm 0.027$ | $0.694 \pm 0.020$ |
| LDS | $0.831 \pm 0.017$ | $0.727 \pm 0.021$ |
| NeuralSparse-GCN | $0.821 \pm 0.014$ | $0.724 \pm 0.014$ |

Table S5: Node classification performance in setting C

|  | Cora + kNN | Citeseer + kNN |
|---|---|---|
| GCN | $0.631 \pm 0.014$ | $0.646 \pm 0.009$ |
| LDS | $0.731 \pm 0.019$ | $0.725 \pm 0.013$ |
| NeuralSparse-GCN | $0.751 \pm 0.013$ | $0.743 \pm 0.007$ |

A and Setting C. LDS performs slightly better in Setting B. From the above observation, we conjecture that NeuralSparse is more robust to graphs with more random edges while LDS is more suitable in a graph of relatively less noise by adding additional edges. We will verify the conjecture in the next subsection.

## S6.2 RANDOM EDGE ADDITION TO CORA AND CITESEER

We further compare NeuralSparse and LDS (Franceschi et al., 2019) on the node classification tasks where original graph structure is available but more random edges are introduced as noise. Starting from the original graphs, we add edges by randomly sampling two nodes $u$, $v$ from node set $\mathbb{V}$ and connecting them.

The results are shown in Figure S2. In both datasets, NeuralSparse achieves better performance compared with LDS as the noise level goes beyond 200%. When the amount of noise increases, the classification accuracy of LDS drops significantly.

This result confirms our conjecture that NeuralSparse is more robust to random edges, compared to LDS.

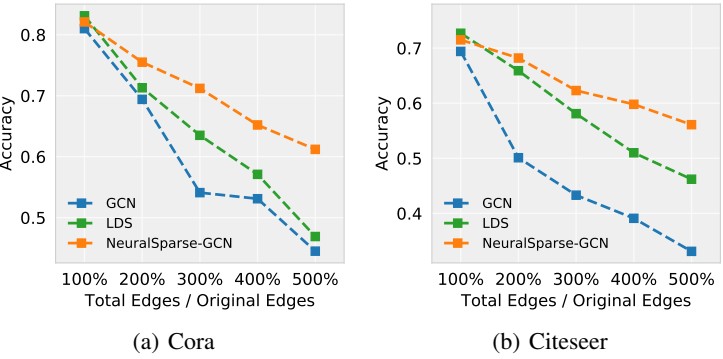

(a) Cora          (b) Citeseer

Figure S2: Node classification performance when adding noise to graph structure.

With the above comments and experimental results, we hope to clarify the difference between the two models and demonstrate that our proposed NeuralSparse is more robust to noises in real-life graphs.

## S7  HOW DO TASK-IRRELEVANT EDGES COULD NEGATIVELY IMPACT THE PERFORMANCE OF GCN?

In this section, we use an example to demonstrate how an input graph with task-irrelevant edges could impact the performance of GCN. For the ease of discussion and visualization, we focus on a GCN with a simple architecture and synthetic graphs where we could adjust graph topology by hyper-parameters in graph generators.

In terms of GCN, we investigate a one-layer GCN

$$f_W = Softmax(\hat{A}XW) = Softmax(ZW) \tag{7}$$

where $\hat{A}$ is a normalized adjacency matrix, $X$ is the input node feature matrix, $W$ is the GCN parameters, and $Z = \hat{A}X$ denotes node representations in the aggregation space. Intuitively, the quality of $Z$ has direct impact to this GCN's performance.

In terms of input graphs, we generate them for node classification tasks as follows.

1. **Nodes and their labels**. 2,000 nodes are generated, where 1,000 nodes are assigned with positive labels and the rest are assigned with negative labels.

2. **Node features**. Each node has a two-dimensional feature vector. For positive nodes, the node features are generated from a Gaussian distribution, where $\mu_{pos} = (-0.5, 0)$ and $\Sigma_{pos}$ is a diagonal matrix with $\Sigma_{pos}[0, 0] = \Sigma_{pos}[1, 1] = 0.3$. For negative nodes, the node features are generated from another Gaussian distribution, where $\mu_{neg} = (0.5, 0)$ and $\Sigma_{neg}$ is a diagonal matrix with $\Sigma_{neg}[0, 0] = \Sigma_{neg}[1, 1] = 0.3$.

3. **Edges**. Given a hyper-parameter $\bar{d}$, for each node, it randomly selects $\bar{d}$ nodes as its one-hop neighbors. With respect to this node classification task, an edge that connects two nodes of different labels could be irrelevant, bringing noise to the GCN.

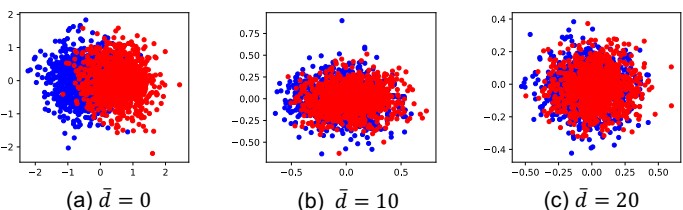

Figure S3: Distributions of $Z$ in graphs with different $\bar{d}$

In Figure S3, the distributions of node representation $Z$ are demonstrated at different $\bar{d}$. When $\bar{d}$ is 0, $\hat{A}$ is an identity matrix so that we simply use input node features in model learning. As shown in Figure S3(a), it is difficult to find a good boundary that well separates the positive and negative nodes by using node features only. However, when we adjusts $\bar{d}$ to 10 or 20 with richer connections, the situation doesn't get better. Because of the noise introduced by irrelevant edges, it becomes harder to find the classification boundary. While a deep learning may still be able to find a complex boundary that well separates the training data, the boundary could overfit the introduced noise, resulting in low generalization power.

In Figure S4, we illustrate how NeuralSparse enhances the prediction accuracy of the GCN. In particular, we focus on the graph with $\bar{d} = 20$, and sparsify this graph by NeuralSparse. As shown in Figure S4, ranging the hyper-parameter $k$ from 1 to 15, the distributions of $Z$ vary. When $k$ is 1, there is no significant change compared with the distribution in Figure S3(a), as the amount of accessible relevant graph data is still small. When $k$ is increased to 5 or 10, the classification boundary becomes much clearer. As the edge generation process is uniformly random, the expected number of relevant edges per node is roughly 10. When $k$ is 15, this $k$ could be an over-estimate on the amount of relevant edges, making it a bit harder to find a good separation.

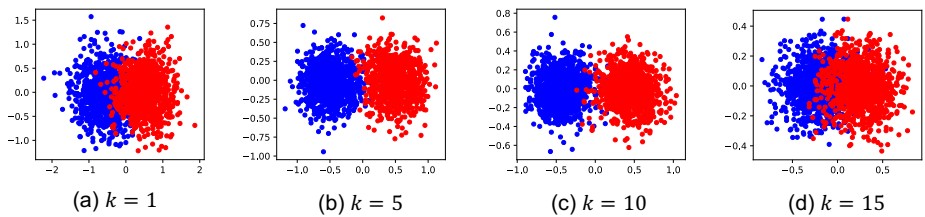

(a) $k = 1$      (b) $k = 5$      (c) $k = 10$      (d) $k = 15$

Figure S4: Distributions of Z in sparsified subgraphs by NeuralSparse

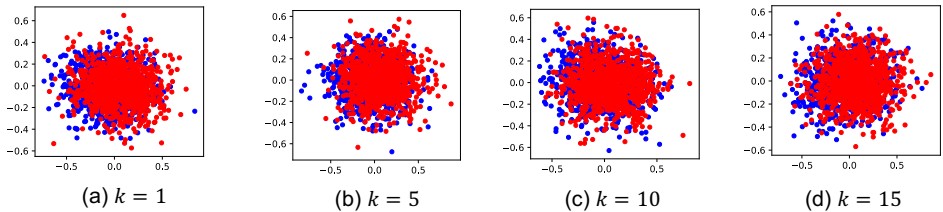

(a) $k = 1$      (b) $k = 5$      (c) $k = 10$      (d) $k = 15$

Figure S5: Distributions of Z in sparsified subgraphs by random downsampling

In Figure S5, we demonstrate how random downsampling could impact the prediction accuracy of the GCN. In general, we could not see any significant improvement. Indeed, it is crucial to perform a task-driven sparsification as NeuralSparse does.

