# OpenReview forum: "Robust Graph Representation Learning via Neural Sparsification"
_ICLR.cc/2020/Conference — Reject_

### Official Review · AnonReviewer3 · 2019-10-18
**Official Blind Review #3**

**Rating:** 6

**Review:**

The authors propose a supervised graph sparsification technique that "mitigates the overfitting risk by reducing the
complexity of input graphs."

The idea is as follows: there is a sparsification network which samples subgraphs (adjacency matrices) by computing a probability distribution for each edge and drawing the existence of each edge from this distribution. The sparsified graph is then fed to a GNN that computes a classification loss. Since the authors use the Gumbel-softmax trick the method is end-to-end differentiable and the output consists of a node classifier and a graph generative model that can be used to sample sparsified graphs.

The paper covers an interesting topic and results in some good numbers on standard benchmark datasets. I also like the idea to use the Gumbel-softmax trick to make the entire model differentiable.

Unfortunately, the authors miss to cite and discuss highly related work [1] (ICML 2019). In this work, the authors also maintain a graph generative model (also by parameterizing edges but with iid Bernoulli RVs), also sampling graphs from this generative model, and also using these sampled graphs to train a GNN for semi-supervised node classification. The resulting model is also end-to-end differentiable. Instead of using a Gumbel-softmax to keep the method differentiable (given that we have discrete random variables) the authors propose a novel way to compute gradients for the parameters of the Bernoulli RVs by posing the problem as a bilevel optimization problem. In [1] the graph cannot only be sparsified but also enriched with edges that might be beneficial for the classification accuracy. Indeed, it was shown that adding edges is more beneficial than removing edges. The results on Cora and Citeseer are better than the results reported by the authors in this submission. At the very least, the authors should familiarize themselves, discuss, and compare empirically to [1].

The existence of this previous work also reduces the novelty of the proposed approach. However, if a  comparison to [1] would be added, the submission could be seen as an alternative instance of the framework presented in [1]. I would be willing to increase my score to accept, if the authors provide such a comparison in an updated version.


[1] https://arxiv.org/abs/1903.11960


-----

I've read the rebuttal and while I don't agree necessarily with all statements made by the authors, I am happy to increase my score based on the discussion of previous work.

**Experience Assessment:**

I have published in this field for several years.

**Review Assessment: Checking Correctness Of Derivations And Theory:**

I assessed the sensibility of the derivations and theory.

**Review Assessment: Checking Correctness Of Experiments:**

I assessed the sensibility of the experiments.

**Review Assessment: Thoroughness In Paper Reading:**

I read the paper thoroughly.

---

> ### Author Response · Authors · 2019-11-09
> **Response to the comments from the reviewer**
>
> Thanks for your valuable comments to our paper. We detail our response to your questions or concerns as follows.
>
> Question 1: What is the difference between NeuralSparse and LDS [1]?
>
> Thanks for sharing this related work [1]. While NeuralSparse and LDS do share the spirit of sampling adjacency matrix for downstream graph learning tasks, NeuralSparse is significantly different from LDS in the following aspects.
>
> First, the assumptions behind NeuralSparse and LDS are different, pursuing different output graph samples.
> - In NeuralSparse, we assume the input graphs are complete, but not all the input graph data are relevant to the downstream tasks. Therefore, NeuralSparse samples subgraphs of input graphs. In other words, graph sampling under NeuralSparse is constrained by input graphs, and the resulting graph samples are strictly subgraphs of input graphs.
> - In LDS, the assumption is the input graphs are incomplete. Therefore, the expected graph samples are subgraphs of a complete graph with respect to given nodes. In other words, there is no guarantee that graph samples are subgraphs of input graphs in LDS.
>
> Second, NeuralSparse is an inductive technique that is capable of handling both inductive and transductive tasks, while LDS is a transductive method that focuses on transductive tasks.
> - In NeuralSparse, the probability of an edge to be sampled is conditioned on node and edge features. In this way, a learned NeuralSparse can directly adapt to unseen testing data in an inductive setting, and the number of parameters is independent of input graph size.
> - In LDS, each possible edge is associated with a parameter. A learned LDS is expected to only work for its input training graph. Let n be the number of nodes in an input graph. The number of parameters in LDS scales quadratically with respect to n.
> As NeuralSparse is an inductive method, it can handle both inductive and transductive tasks
>
> In addition, NeuralSparse has better scalability. Given a fixed hyper-parameter k, NeuralSparse scales linearly with respect to the number of edges in an input graph. Meanwhile, LDS is difficult to scale with a large number of nodes, even if its underlying graph is sparse.
>
> In the latest draft, we have added the discussion on [1], in Section 2 and the end of Section 4.
>
> Question 2: Empirical comparison between NeuralSparse and LDS.
>
> In the latest draft, we have added S6 in the appendix for a detailed discussion on the empirical comparison between NeuralSparse and LDS. The following is a brief summary.
>
> [Datasets]
> As LDS is transductive only and the tasks in Reddit, PPI, as well as Transaction datasets are inductive, we focus on Cora and Citeseer in this set of experiments.
>
> [Input graphs]
> LDS utilizes edges in an input graph to initialize edge parameters. In general, there are three possible ways to prepare input graphs.
> - Setting A: k-NN graphs [1]
> - Setting B: original input graphs
> - Setting C: edge union of k-NN graphs and original input graphs
>
> [Key observations]
> In general, NeuralSparse and LDS achieve comparable prediction accuracy.
> - NeuralSparse performs slightly better in Setting A and Setting C
> - LDS performs slightly better in Setting B.
>
> By comparing the original graphs and k-NN graphs in terms of the percentage of edges connecting nodes of same labels, we find k-NN graphs include significantly more irrelevant edges with respect to this task as shown below.
> +-------------+---------+-------------+
> |                 | Cora  | Citeseer  |
> +-------------+---------+-------------+
> | Original  | 82.2%| 73.1%     |
> +-------------+---------+-------------+
> | k-NN       | 53.9%| 48.4%      |
> +-------------+---------+-------------+
>
> We conjecture that NeuralSparse is more robust to graphs with more random edges (e.g., Setting A and C), while LDS could perform better in a graph of relatively less noise (e.g., Setting B) by adding additional edges.
>
> [Robustness against random edges]
>
> To confirm NeuralSparse is more robust against irrelevant edges, we evaluate GCN, LDS-GCN, and NeuralSparse-GCN on original graphs plus additional random edges. The detailed data preparation is described in S6.
>
> On both Cora and Citeseer datasets, we observe similar trends. First, when we increase the amount of random edges, the accuracy of the three methods decreases. Second, starting from 200%, NeuralSparse-GCN consistently performs the best in all the cases. This suggests NeuralSparse-GCN is more robust against random (potentially irrelevant) edges, compared with the other two methods.
>
> Reference
> [1] Franceschi, Luca, et al. "Learning discrete structures for graph neural networks." ICML 2019.

---

> > ### Comment · AnonReviewer3 · 2019-11-15
> > **Response to rebuttal**
> >
> > Thank you for your constructive rebuttal. Due to the newly added comparison and discussion of previous work, I have increased my score.
> >
> > The reasons why I still think that it is not a strong accept is that it is in the end a method very similar to previous ones (either GAT --> which, intuitively, also learns which edges to keep) and [1] which is able to truly remove (and add) edges in a discrete, non-continuous manner.

---

### Official Review · AnonReviewer2 · 2019-10-22
**Official Blind Review #2**

**Rating:** 1

**Review:**

The authors argue that existing GCN-based approaches may pose non-trival overfitting risk during the training phase, especially when high-dimensional features and high-degree entities are observed in the graphs. To address the issue, the authors integrate graph sparsification with conventional graph neural nets. Experimental results show the efficacy of the proposed model in a series of benchmark datasets. In general, the paper is easy-to-follow and well-organized. My main concern is there lack some insightful discussion regarding the problem motivation and the proposed algorithm. In particular,
(1) It is unclear why existing GCN-based approaches can not handle the cases shown in Fig. 1. Is there any evidence (either theoretical or empirical) or reference to support this argument?
(2) The motivation example shown in Fig. 1 is confusing. Conventionally, graph sparsification aims to find smaller subgraphs from the input graphs that preserve the key structures. However, in Fig. 1 (b), the sparsified subgraph seems only downsampling the edges while preserving all the nodes as the original graph. The authors may want to clarify whether the sparisified subgraph has the identical size as the input graph.
(3) Some notations are not formally defined before using them. In Eq. 2, what do Q_\theta and Q_\phi denote?
(4) The statement of "trade-off between model accuracy and graph complexity by tuning the hyperparameter k" is vulnerable. If the overfitting exists, larger k may result in lower accuracy in the testing phase.
(5) What is the complexity of f_\Phi()?
(6) The complexity (i.e., f(km)) of the proposed model is problematic. As stated at the beginning of this paper, the paper targets the graph with "complex local neighborhood", where each node is described by rich features and neighbors. In other words, the target graph is not sparse. In this case,  the complexity of the proposed algorithm can be intractable, especially when k is large and m is close to n^2.

**Experience Assessment:**

I have published in this field for several years.

**Review Assessment: Checking Correctness Of Derivations And Theory:**

I carefully checked the derivations and theory.

**Review Assessment: Checking Correctness Of Experiments:**

I carefully checked the experiments.

**Review Assessment: Thoroughness In Paper Reading:**

I read the paper thoroughly.

---

> ### Author Response · Authors · 2019-11-09
> **Response to the comments from the reviewer**
>
> Thanks for your valuable comments to our paper. Our answers to your questions/concerns are detailed as follows.
>
> Question 1: It is unclear why existing GCN-based approaches can not handle the cases shown in Fig. 1. Is there any evidence (either theoretical or empirical) or reference to support this argument?
>
> Thanks for the suggestion. In the latest draft, we have added S7 in the appendix where we use an example to illustrate one of the scenarios where GCN could suffer sub-optimal performance by directly processing original input graphs.
>
> Question 2: Conventionally, graph sparsification aims to find smaller subgraphs from the input graphs that preserve the key structures. However, in Fig. 1 (b), the sparsified subgraph seems only downsampling the edges while preserving all the nodes as the original graph. The authors may want to clarify whether the sparisified subgraph has the identical size as the input graph.
>
> In terms of node size, yes, a sparsified subgraph has the same number of nodes as its original graph does. In terms of edge size, no, for each node, it has no more than k edges, given a fixed hyper-parameter k. We have updated the definition of k-neighbor subgraphs in Section 4 to clear possible confusion.
>
> Question 3:  Some notations are not formally defined before using them. In Eq. 2, what do Q_\theta and Q_\phi denote?
>
> Q_\theta is the approximation function for P(Y|g) and Q_\phi is the approximation function for P(g|G). We have updated in the latest draft accordingly to clear such confusion.
>
> Question 4: The statement of "trade-off between model accuracy and graph complexity by tuning the hyper-parameter k" is vulnerable. If the overfitting exists, larger k may result in lower accuracy in the testing phase.
>
> Thanks for pointing out this confusion. In the latest draft, we have updated the relevant discussion in Section 4. In particular, the quoted statement is modified as "We are able to adjust the estimation on the amount of task-relevant graph data by tuning the hyper-parameter k."
>
> Question 5: What is the complexity of f_\Phi()?
>
> f_\phi() is implemented by a multi-layer perceptron (MLP) with Gumbel-Softmax. Let $d_n$ and $d_e$ be the numbers of dimensions for node and edge features, respectively. The MLP is implemented in a two-layer architecture with hidden dimensionality d_h. The complexity of the MLP is $O(d_h * (2d_n + d_e))$. Suppose the number of one-hop neighbors of a node is $\hat{d}$. This function visits one node's one-hop neighborhood in $O(d_h * (2d_n + d_e) * \hat{d})$.
>
> Question 6: The complexity (i.e., O(km)) of the proposed model is problematic. As stated at the beginning of this paper, the paper targets the graph with "complex local neighborhood", where each node is described by rich features and neighbors. In other words, the target graph is not sparse. In this case, the complexity of the proposed algorithm can be intractable, especially when k is large and m is close to $n^2$.
>
> We answer this question from the following aspects.
>
> - Does NeuralSparse require or expect dense graphs? As discussed in S7, even for a graph with 1000 nodes and average degree 20, GCN still could suffer sub-optimal performance by processing irrelevant edges. Therefore, as long as there are irrelevant edges that impact the prediction performance in the downstream tasks, NeuralSparse will be effective. In sum, NeuralSparse has no assumption or expectation to the density of input graphs.
>
> - When k is large and m is close to $n^2$, is NeuralSparse intractable? First of all, k is a hyper-parameter in NeuralSparse and independent to input graph size. Given a fixed k, NeuralSparse scales linearly with respect to m. In practice, k linearly impacts computation time, and one can tune k in order to meet computation resource constraints. Second, if m is large, downstream GNNs will face the same scalability challenge. Fortunately, similar to existing GNNs, NeuralSparse also adopts independent node-centric computation, which is highly parallelizable as suggested in Section 4. This could address the scalability issue to some extent. As the scalability challenge is a common problem to existing GNNs and NeuralSparse, a comprehensive discussion of this problem could be out of our scope. In sum, we argue that the extra computation overhead from NeuralSparse is tunable and affordable, compared with the complexity in downstream GNNs.

---

> ### Author Response · Authors · 2019-11-14
> **Appreciate your time and attention**
>
> Dear Reviewer,
>
> If you have more questions or concerns about our latest draft or response, please feel free to let us know. We are happy to discuss with you.

---

### Official Review · AnonReviewer1 · 2019-10-25
**Official Blind Review #1**

**Rating:** 8

**Review:**

The paper proposes a trainable graph sparsification mechanism that can be used in conjunction with GNNs. This process is parameterized using a neural network and can be trained end-to-end (by using the Gumbel softmax reparameterization trick) with the loss given by the task at hand. Experimental results on node classification tasks are presented.

The paper is well written and easy to follow. I think that overall the method is sound and well executed.

Empirical results are convincing as the method consistently improves performance (compared to not applying the sparsification) over several GNNs. I feel that the claim of the improvement could be softened a bit. Please correct me if I'm reading Table 2 wrongly, with the exception of the Transaction dataset, most differences are below 3%. I am not familiar with these datasets so I cannot judge the significance of the improvement. In any case, the reduction in computation with the improved performance is a strong result.

The baselines that include unsupervised graph sparsification as a pre-processing make the results worse (with respect to not applying it) in all cases. This shows that for this problem, a task driven specification is crucial for maintaining performance.

Neural Sparse model has more parameters than the version that does not use a sparsifier. Do you think that this could influence the performance? Would it be possible to compare using similar number of trainable parameters? I assume that using more parameters would not imply better performance for the baseline, but it would be good to clarify.

I can understand that the performance of the model depends critically on k, and that k might vary significantly over datasets. In my view, it would be informative to include (maybe in the supplementary material) the performance variations on the corresponding validation sets as one changes k for the different datasets (as done in Figure 3 (c)).

In Algorithm 2 it would be better to use a different letter for the edge set (as currently looks like the real numbers).

**Experience Assessment:**

I have read many papers in this area.

**Review Assessment: Checking Correctness Of Derivations And Theory:**

I assessed the sensibility of the derivations and theory.

**Review Assessment: Checking Correctness Of Experiments:**

I assessed the sensibility of the experiments.

**Review Assessment: Thoroughness In Paper Reading:**

I read the paper at least twice and used my best judgement in assessing the paper.

---

> ### Author Response · Authors · 2019-11-09
> **Response to the comments from the reviewer**
>
> Thanks for your valuable comments to our paper.
>
> For the comments on performance results in Table 2, we agree with you. As you pointed out, our empirical results suggest NeuralSparse is able to help existing GNN techniques achieve comparable or even better generalization performance by effectively reducing input graph complexity and removing graph noise. In addition, we have updated our analysis with respect to the performance results in Table 2 accordingly.
>
> For the impact brought by a total number of trainable parameters, we have added S4 in the appendix to detail the discussion. In particular, we focus on GCN in this set of experiment, and have added a baseline called NeuralSparse-GCN-Compact which shares a similar number of trainable parameters with an original GCN. We summarize the empirical results in Table S2 as follows.
> - Both NeuralSparse-GCN-Compact and NeuralSparse-GCN consistently outperform GCN across all the datasets.
> - Compared with NeuralSparse-GCN, NeuralSparse-GCN-Compact achieves comparable prediction accuracy with smaller variance in most cases.
>
> For the question of how the hyper-parameter k impacts validation performance, we have added S5 in the appendix for the discussion. Our observation is summarized as follows. In terms of validation, as shown in Figure S1, the validation performance increases when k ranges from 2 to 10 with more available graph data.  After k exceeds 10, the increase in validation performance slows down and turns to be saturated or dropping.  In terms of testing performance, it shares a similar trend when k ranges from 2 to 10. Meanwhile, the testing performance drops more after k exceeds 10.
>
> In Algorithm 2, the letter for the edge set has been updated with $\mathbb{H}$ to avoid confusion.

---

### Author Response · Authors · 2019-11-15
**Appreciate your time and discussion in rebuttal**

Dear AC and reviewers,

We sincerely appreciate the valuable comments from the reviewers. The constructive suggestions indeed help us further improve the paper.

Meanwhile, we feel sorry for not receiving further feedback from reviewer#2. We fully respect the reviewer's comments and have spent significant effort in addressing the concerns. In the latest draft and response posted on 11/09, we believe we have addressed the concerns from reviewer#2. If there are further questions and concerns, we are eager for discussion. However, we could not find a response or updated rating from reviewer#2. If this is the final evaluation, it is unfair for us and impairs the trust between authors and reviewers in ICLR.

In sum, we hope the decision process could cautiously consider the reviewer#2’s comments, and give our paper a fair evaluation.

Thanks.

---

### Decision · Program_Chairs · 2019-12-19

**Decision:**

Reject

**Comment:**

This submission proposes a graph sparsification mechanism that can be used when training GNNs.

Strengths:
-The paper is easy to follow.
-The proposed method is sound and effective.

Weaknesses:
-The novelty is limited.

Given the limited novelty and the number of strong submissions to ICLR, this submission, while promising, does not meet the bar for acceptance.